# Simulating Human Visual Perception in Tunnel Portals

Changjiang Liu [1,2,*] and Qiuping Wang [1]

1   School of Civil Engineering, Xi'an University of Architecture & Technology, Xi'an 710055, China;
    wangqiuping1962@foxmail.com
2   Shaanxi Province Transport Planning Design and Research Institute, Xian 710065, China
*   Correspondence: lcj2012@foxmail.com; Tel.: +86-02968718767

**Abstract:** To study the characteristics of light and dark adaptation in tunnel portals, and to determine the influencing factors in light–dark vision adaptation, basic tunnel lighting and linear design data were obtained. In this study, we used a light-shielded tent to simulate the dark environment of a tunnel, observe the driver recognition time for target objects during the light–dark adaptation process, and analyze the light–dark adaptation time of human vision. Based on the experimental data, we examined the relationships between age, gender, illuminance, and light and dark adaptation times, and established a model for these relationships. The experimental results show that the dark adaptation time is generally longer than the light adaptation time. The dark adaptation time is positively related to age and exhibits a cubic relationship. There is no significant correlation between the light adaptation time and age, but the overall trend is for the light adaptation time to gradually increase with increasing age. There is no correlation between gender and light and dark adaptation times, but there is a notable correlation between light and dark adaptation times and illuminance. When the illuminance ranges from 11,000 to 13,000 lux, the light and dark adaptation times are the longest.

**Keywords:** vision; human visual; perception; tunnel portal; light adaptation; dark adaptation; adaptation time

## 1. Introduction

Tunnel entrance and exit sections are transition sections in a vehicle operating environment. When driving from the natural light environment of a highway into the artificial light environment of a tunnel entrance section, or when exiting from the artificial light environment of a tunnel exit section, the highway produces different environmental natural light conditions. When the differences are stark, these can lead to a visual lag in drivers and can cause short-term visual impairment. This is often called the black hole or white hole effect [1]. It can affect driver recognition of vehicles or obstacles ahead, driving stability, and safety and comfort, which can easily lead to increased stress on drivers, operational errors, and traffic accidents.

Research on the visual characteristics of tunnel entrance and exit sections has been a topic of considerable interest in the field of tunnel engineering. Scholars have conducted extensive research in this field, but few meaningful results have been obtained.

In the study of driver visual characteristics in tunnel entrance and exit sections, Narisada et al. [2] used eye tracker records to establish an index of gaze characteristics to describe eye movement attributes. Zwahlen et al. [3] used an onboard video eye movement recording system to record various eye movement data for drivers at tunnel entrances and analyzed the visual parameters of gaze and saccade characteristics. Maltz et al. [4] studied eye movement characteristics in the visual search process of drivers, the results suggested how aging might affect the efficacy of visual information processing. Crundal et al. examined the average gaze duration, number of gaze points, and other related eye movement characteristics of drivers based on observation of the visual search process

of drivers in different road environments [5,6]. Ito et al. [7] established a model of the relationship between driver adaptation time and road brightness based on the adaptation time from when a driver notices a reaction target until the time when its outline is discerned. Liu [8] used the heart-rate change index to evaluate the impact of tunnel lighting on drivers and established a relationship model between the heart-rate change and influencing factors. Du et al. [9,10] proposed the maximum transient speed of the pupil area (MTPA) as an indicator of the visual load in response to visual shocks in light and dark adaptation. Hu [11] et al. implemented a point–point brightness meter to measure the brightness at the entrance and exit of a tunnel and used the equivalent light curtain brightness to calculate the adaptive brightness at each location. In addition, a mathematical model of the relationship between the brightness levels at the tunnel entrance and outside the tunnel was established [12]. Hu et al. [13] calculated the threshold value of the brightness difference in the entrance section of a tunnel at night at different design speeds. Huang [14] found that when entering a tunnel, an excessively high illuminance transition rate tends to increase the driver's visual load and psychological tension. Mehri et al. [15] compared the safety lighting level at a tunnel entrance with the de Boer scale and found that the black hole effect can cause the eyes of a driver to not readily adapt to brightness level changes at the tunnel entrance, thereby increasing the risk of traffic accidents along this section. Chen et al. reported that the optimum dark adaptation position of the driver of a large truck is closer to the door, and there is a higher visual load when driving along a tunnel entrance section, resulting in poor driving safety [16–19]. Zhaoet al. analyzed the influence of light-emitting diode (LED) color rendering on the dark adaptation of the human eye at a tunnel entrance [20]. Dong et al. [21] revealed that LEDs with correlated color temperatures (CCTs) ranging from 4000 to 4500 K are preferable for tunnel entrance lighting. Zhi et al. [22] evaluated the safety of illuminance transitions at highway tunnel portals on the basis of visual load. Zhou et al. [23] simulated human visual perception under nighttime illuminance.

In the study of the relationship between human visual characteristics and age, previous results [24] demonstrate that visual characteristics degenerate with age and the ability to adjust the eye lens, pupil agility, visual sensitivity and speed and total amount of dark and light adaptation all decrease. The final effect of age on vision is visual acuity deterioration. The highest visual acuity occurs at an age between 14 and 20 years. After the age of 30 years, visual acuity gradually decreases. Beyond the age of 60 years, visual acuity is only 1/3–1/4 of that at an age of 20 years. The research of Lokka et al. [25] showed that an important influencing factor of visual spatial memory. Keil et al. [26] believed that after visual training, cognitive function can be improved.

In a dynamic road traffic system consisting of people, cars, roads, and different environments, more than 80% of road, traffic, and environmental information during driving is obtained by drivers through vision [27]. According to the aforementioned studies, the light and dark changes in a tunnel portal affect human visual performance [28]. However, quantitative evaluation of the light and dark adaptation characteristics of drivers is uncommon.

In this study, a light-shielded tent was used to simulate the dark environment in a tunnel. Through a large number of observations, the actual measurement of the driver's adaptation time under different illuminations was obtained. We analyzed the light and dark adaptation characteristics of drivers when they entered and left the tunnel, and studied the effects of gender, age, and illumination.

## 2. Materials and Methods

### 2.1. Experimental Elements

In the experiments, a light-shielded tent was used to simulate the dark environment of a tunnel. The experimental site was located in an open campus grassland. In addition, to simulate the noise environment of a tunnel portal, the noise was controlled at 50–55 dB [29].

To ensure that the illuminance at the simulated tunnel entrance or exit changed significantly, the experiments were conducted on a sunny day, between 09:30 and 19:30.

### 2.2. Observers

People who had been driving for longer than 1 year were chosen for the experiments, and they included 5 age groups: 21–30, 31–40, 41–50, 51–60, and 61–70 years. Across the different age groups, male and female drivers were distributed as equally as possible. All observers had normal vision or a normal corrected vision. None of the observers suffered from night or color blindness or other conditions affecting night vision performance. All observers were well rested and exhibited normal reactions, and alcohol consumption and medication usage were prohibited during the experimental period.

To ensure the accuracy of our experimental parameters, it was necessary to adopt a certain sample size. Taking the measured adaptation time as an accuracy parameter, the minimum measured sample size was calculated as follows:

$$N \geq \left(\frac{\sigma K}{E}\right)^2, \tag{1}$$

where $N$ is the sample size; $\sigma$ is the overall standard deviation, and a value of 1 s was adopted for this study; $K$ indicates the confidence level, and when a confidence level of 95% is considered, $K = 1.96$; $E$ is the allowable error, which was 0.15 s for this study.

Based on Equation (1), $N \geq 171$ was obtained. In this study, 237 observers were chosen (see Figure 1).

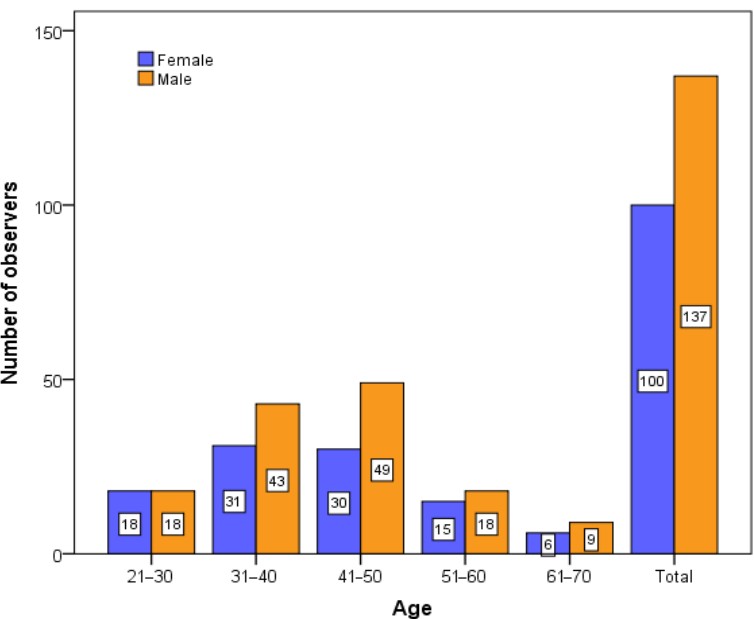

**Figure 1.** Composition of the observers.

### 2.3. Experimental Setup

1.  Light-shielded tent (see Figure 2a): The dark environment inside a tunnel was simulated with the tent, with a height of 3.0 m, a length of 8.0 m, and a width of 4.0 m. The illuminance ranged from 40~520 lux in the tent.
2.  Digital illuminance meter (see Figure 2b): The ambient illuminance was measured with a TES-1339, and the measurement illuminance range was 0.0–990,000 lux, while the resolution was 0.01 lux. The measurement accuracy was ± 3%.
3.  Camera (see Figure 2c): The adaptation time of the observers was recorded upon entering and leaving the simulated tunnel.

4.  Test board and card: The test board was 1.2 m high. There were 20 test cards, including simple Chinese characters and animal images, with a size of 0.15 × 0.15 m. The distance between the observer and the test card was 60–80 cm, which allowed the graphic elements to be identified.
5.  Car: A noisy environment was simulated.

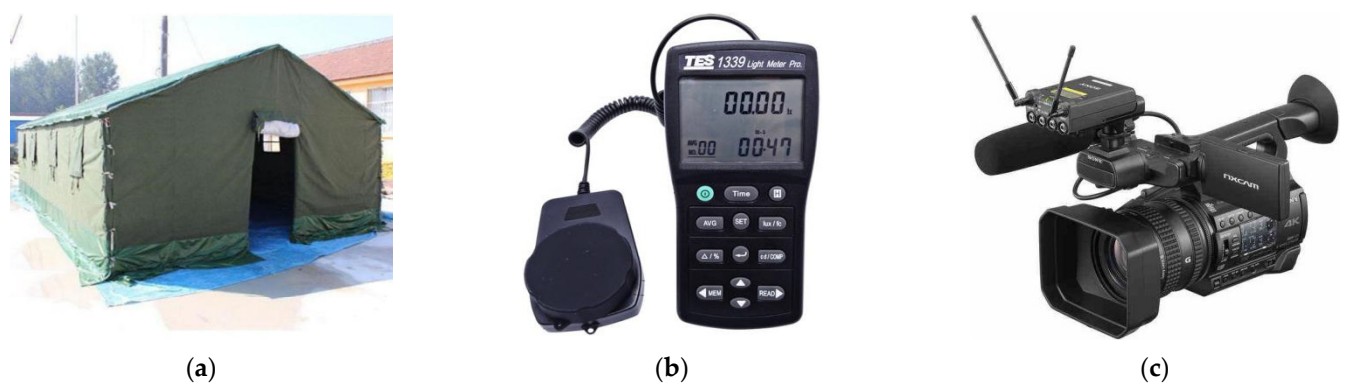

|     |     |     |
| --- | --- | --- |
| (**a**) | (**b**) | (**c**) |

**Figure 2.** Experimental setup: (**a**) light-shielded tent; (**b**) digital illuminance meter; (**c**) camera.

### 2.4. Experimental Procedure

Step 1. Ambient illuminance measurement

The illuminance was measured at 2 m from the entrance side, at the center of the tent, and at 2 m from the exit side of the tent.

Step 2. Dark adaptation time measurement

The observers were notified in advance of the experimental route. First, observers walked into the tent from 2 m outside it and approached the center of the tent, at which point they identified the objects on the test card. A camera recorded the time from entering the tunnel to identifying the objects.

Step 3. Light adaptation time measurement

The observers remained in the tent for 3 min, walked out of the tent, and identified the objects on the test card. Another camera recorded the time from exiting the tent to reading the object names.

Figure 3 shows experimental procedure.

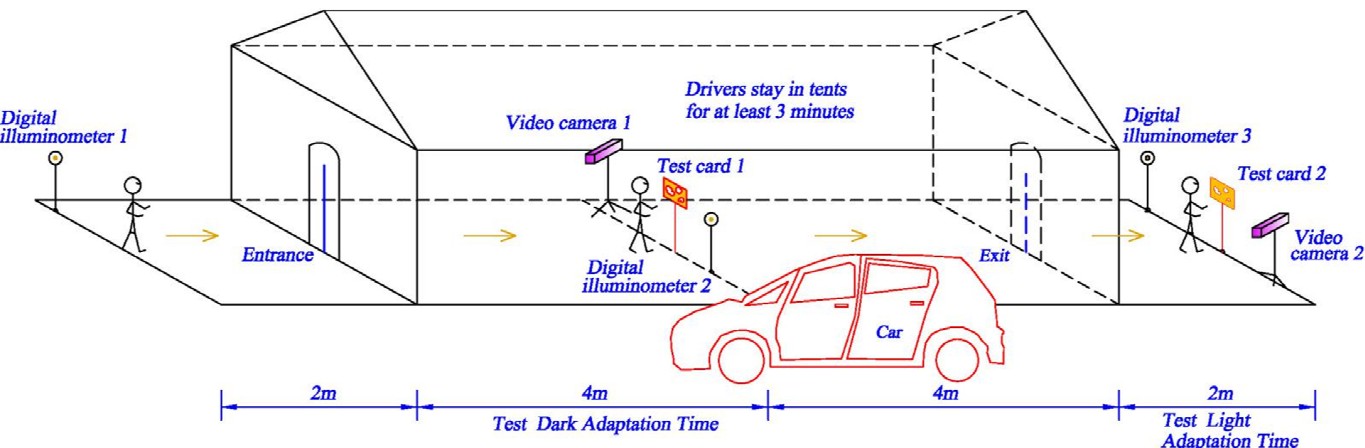

**Figure 3.** Schematic of the experiment.

### 2.5. Experimental Design

**Experiment 1:** A total of 237 observers entered the site randomly for testing, ensuring that the time and personnel were random.

**Experiment 2:** To exclude the effects of age, gender, individual differences, and other factors on the adaptation time, 4 females and 4 males in the age groups 21–30, 31–40, and 41–50 years were randomly selected, for a total of 24 observers. The experiment was repeated at different illuminances, and measurements were recorded every 20–30 min.

### 2.6. Experimental Error Control

To ensure data validity, each experiment was repeated 3 times, and the average data from these three experiments were considered the characteristic visual data of the observers.

## 3. Results and Discussion

### 3.1. Comparative Analysis of the Light and Dark Adaptation Times

Using Experiment 1's data (see Figures 4–6), we compared and analyzed the durations of the light and dark adaptation times.

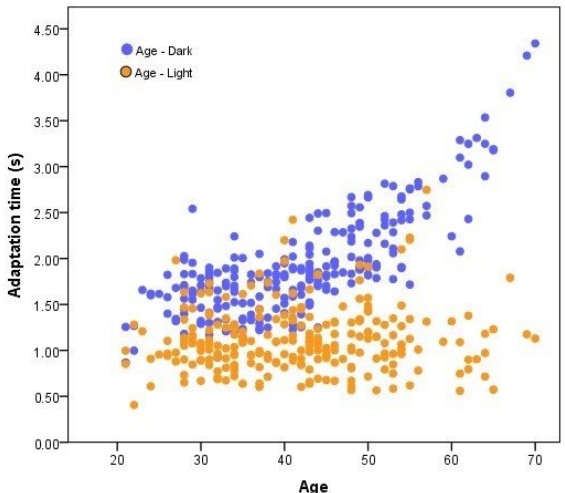

**Figure 4.** Measured light and dark adaptation time distributions.

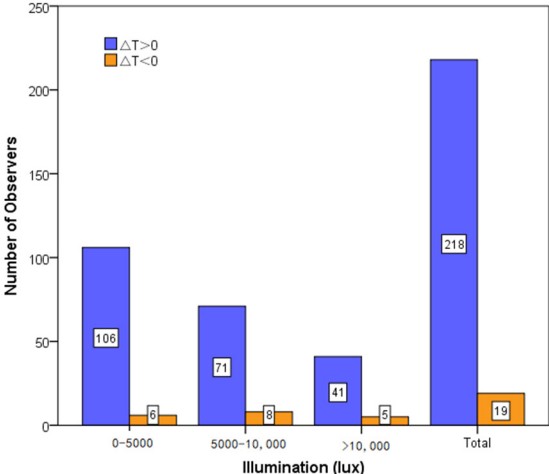

**Figure 5.** Relationship between the adaptation time difference and illuminance.

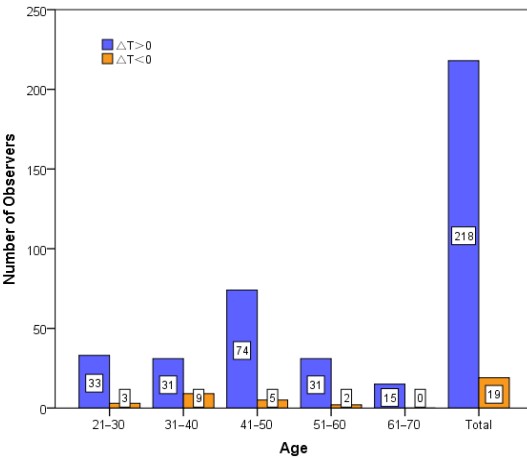

**Figure 6.** Relationship between the adaptation time difference and age.

Denoting the dark adaptation time $T_D$ and the light adaptation time $T_L$, the difference between the dark and light adaptation times is $\Delta T$; then, $\Delta T = T_D - T_L$. In our study, there were 19 cases with $\Delta T < 0$ and 219 cases with $\Delta T > 0$.

Figure 5 shows the relationship between the adaptation time difference and illuminance. Clearly, the dark adaptation time is longer than the bright adaptation time under all the illuminance conditions. This conclusion is the same as in previous research [1,2], which shows that the dark adaptation time is longer than the light adaptation time, indicating that the visual discomfort for drivers caused by the tunnel entrance is more severe than that caused by the tunnel exit; i.e., the "black hole" effect is more dangerous than the "white hole" effect.

Figure 6 shows the relationship between adaptation time difference and age. Clearly, the dark adaptation time is longer than the light adaptation time at all ages.

Figure 7 shows that, as age increases, the difference between the light and dark adaptation times increases. The difference between the dark and light adaptation times is less than 3.0 s, and most of the differences are concentrated between 0.2 and 1.4 s.

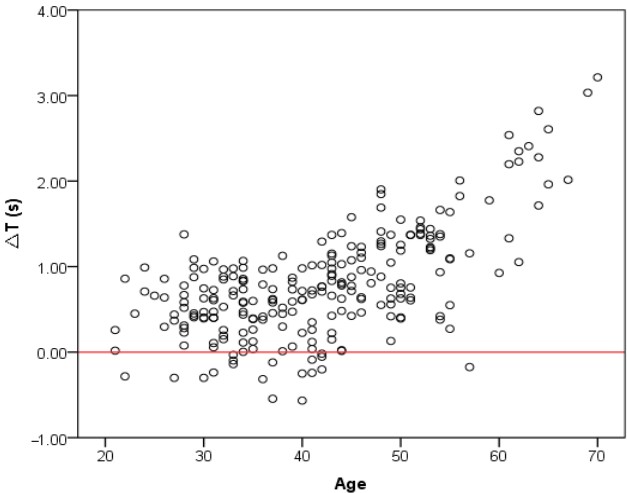

**Figure 7.** Measured adaptation time difference distribution.

### 3.2. Characteristics of Light and Dark Adaptation Based on Age

Based on the Experiment 1 data, we analyzed the light and dark adaptation times at different ages. To reduce the effect of illuminance on the correlation between adaptation time and age, we selected the data at illuminances ranging from 500 to 8000 lux for a total of 187 effective samples, as shown in Figures 8 and 9.

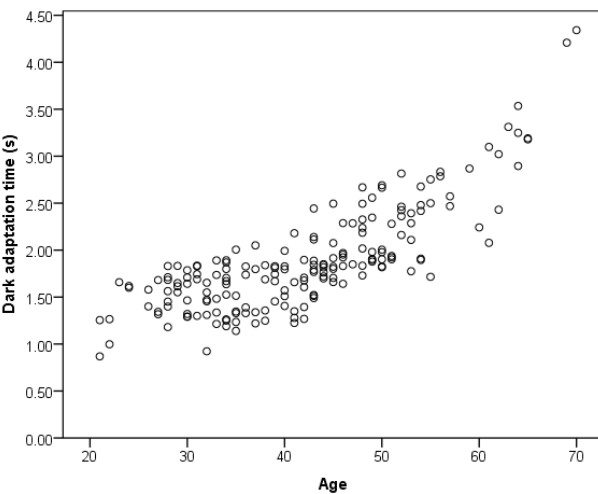

**Figure 8.** Relationship between dark adaptation time and age (500–8000 lux).

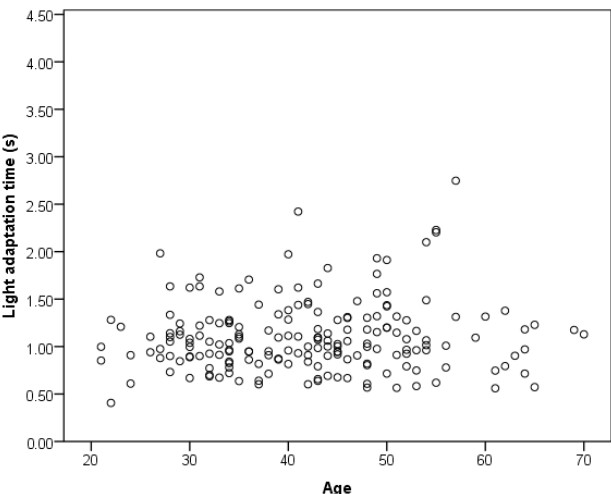

**Figure 9.** Relationship between light adaptation time and age (500–8000 lux).

### 3.2.1. Correlation Analysis for Adaptation Time and Age

We conducted a correlation analysis for dark and light adaptation times and age. The Spearman correlation coefficient is denoted as $R_s$ in Table 1.

**Table 1.** Correlation between adaptation time and age.

| Type | $N$ | Spearman Correlation Coefficient $R_s$ | Significance $p$ |
|------|-----|----------------------------------------|------------------|
| Dark adaptation | 187 | 0.805 [1] | ≤0.001 |
| Light adaptation | 187 | 0.079 | 0.283 |

[1] Correlation is significant at 0.01 (2-tailed).

- The dark adaptation $R_s$ value is 0.805, and the $p$ value is below the theoretical significance level of 0.01, indicating that the dark adaptation time exhibited a significant positive correlation with age.
- The light adaptation $R_s$ value is 0.079, which indicated that there was no significant correlation between the light adaptation time and age.

### 3.2.2. Establishing the Relationship between Dark Adaptation Time and Age

Let age be denoted as *a*, and a regression analysis be performed to obtain Equation (2):

$$T_D = 2.819 \times 10^{-5} a^3 - 0.003 a^2 + 0.092 a + 0.226, \tag{2}$$

Figure 10 shows the obtained model fitting result for the relationship between age and dark adaptation time.

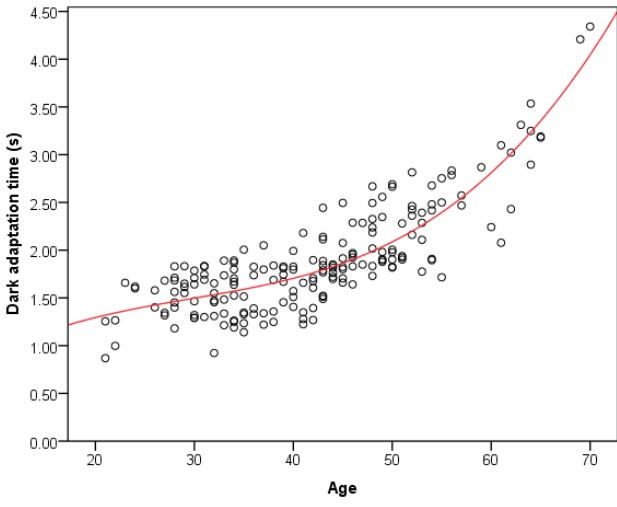

**Figure 10.** Regression curve of dark adaptation time and age.

To validate the model, we performed a correlation test, as summarized in Table 2.

**Table 2.** Model correlation test.

| Model | $R$ | $R^2$ | Adjusted $R^2$ | Standard Error |
|---|---|---|---|---|
| Cubic term | 0.862 | 0.743 | 0.738 | 0.283 |

Table 2 indicates that $R^2$ = 0.738, which implies that the degree of fitting of the cubic function model is acceptable.

The model was further tested with *t* and *F* tests, and the results are listed in Tables 3 and 4, respectively.

**Table 3.** Model *t* test.

| Type | Non-Standardized Coefficients | | Standardized Coefficients | $T$ | Significance *p* |
|---|---|---|---|---|---|
| | $B$ | Standard Error | Beta | | |
| $a$ | 0.092 | 0.067 | 1.780 | 1.386 | 0.167 |
| $a^2$ | −0.003 | 0.002 | −4.210 | −1.626 | 0.106 |
| $a^3$ | $2.819 \times 10^{-5}$ | 0.000 | 3.357 | 2.450 | 0.015 |
| Comments | 0.226 | 0.921 | | 0.245 | 0.807 |

**Table 4.** Model *F* test.

| Type | Sum of Squares | Degrees of Freedom (df) | Mean Squared Error | $F$ | Significance *p* |
|---|---|---|---|---|---|
| Between Groups | 42.297 | 3 | 14.099 | 175.971 | ≤0.001 |
| Within Groups | 14.662 | 183 | 0.080 | | |
| Total | 56.959 | 186 | | | |

Tables 3 and 4 indicate that the significance value $p \leq 0.001$, demonstrating that there is a significant cubic function relationship between age and dark adaptation time.

Our results show the following:

- Dark adaptation time and age generally exhibited a cubic correlation; i.e., the dark adaptation time generally increased with increasing age.
- Equation (2) confirmed statistical significance.
- The dark adaptation time was generally shorter than 4.0 s, and most values were concentrated between 1.2 and 2.4 s.

### 3.2.3. Analysis of the Relationship between Light Adaptation Time and Age

Figure 11 shows the scatter diagram of light adaptation time and age.

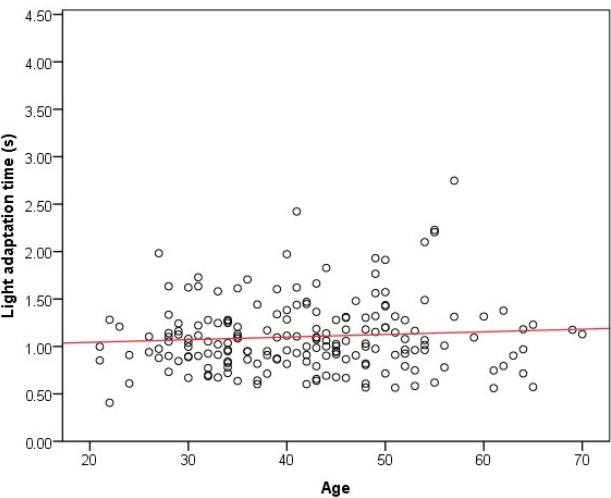

**Figure 11.** Regression curve of light adaptation time and age.

Figure 11 shows the following:

- There were large individual differences in light adaptation time, and there was no clear correlation with age, but the overall trend was a gradual increase with increasing age.
- The light adaptation time was generally shorter than 2.0 s, and most light adaptation times ranged from 0.7 to 1.5 s.

### 3.3. Characteristics of Light and Dark Adaptation Based on Gender

According to the Experiment 1 data, we analyzed the effect of gender on light and dark adaptation times. To eliminate the influences of age and illuminance on the analysis results, data for observers older than 55 years and data obtained at an illuminance exceeding 12,000 lux were excluded. There were 192 effective samples.

Figures 12–15 show the relationships between gender and dark and light adaptation times. Clearly, the adaptation time remained generally the same regardless of gender at different ages and illuminances. Therefore, it can be concluded that there is no correlation between gender and light and dark adaptation times.

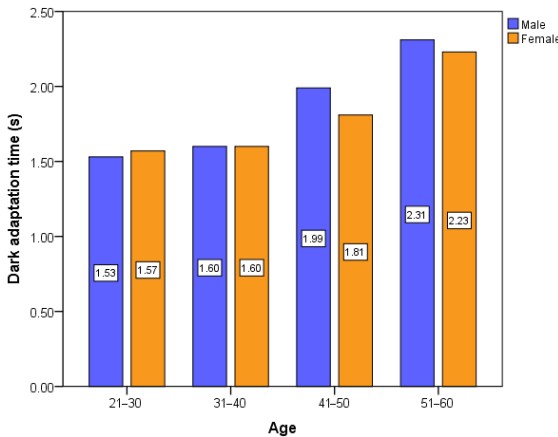

**Figure 12.** Relationship between gender and dark adaptation time by age.

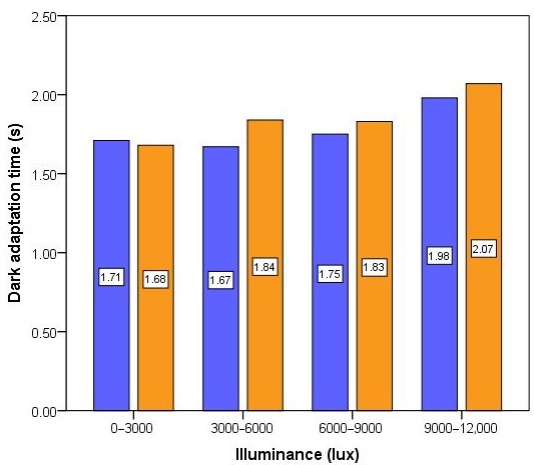

**Figure 13.** Relationship between gender and dark adaptation time by illuminance.

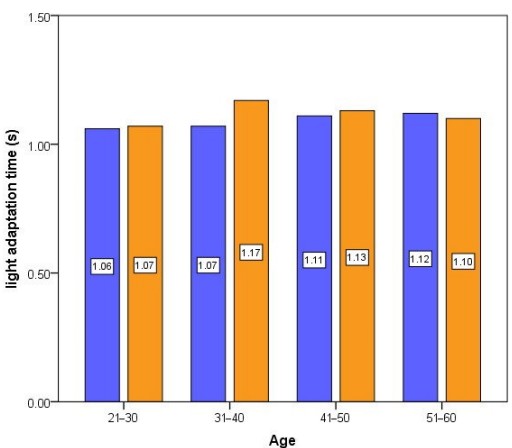

**Figure 14.** Relationship between gender and light adaptation time by age.

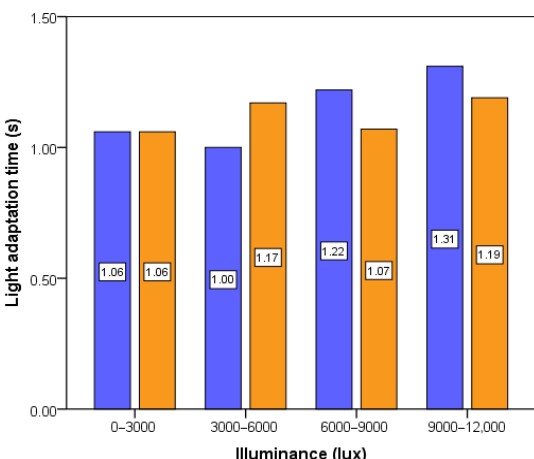

**Figure 15.** Relationship between gender and light adaptation time by illuminance.

*3.4. Characteristics of Light and Dark Adaptation Based on Illuminance*

Using the Experiment 2 data, we analyzed the influence of illuminance on the characteristics of light and dark adaptation responses. In the experimental sample data, the effects of age, gender, and other factors on adaptation time were eliminated, and the effective sample data consisted of 176 groups.

3.4.1. Analysis of the Relationship between Adaptation Time and Illuminance

In this study, box charts were adopted to evaluate the adaptation time at the different illuminances. The statistics at each illuminance level included 5 characteristic metrics: the sample lower boundary, 25th quantile, 75th quantile, sample upper boundary, and average value.

As shown in Figure 16, the dark adaptation time increased with increasing illuminance and reached a maximum value from 11,000 to 12,000 lux. When the illuminance exceeded 11,000–12,000 lux, the dark adaptation time gradually decreased with increasing illuminance, but the downward trend was slower than the upward trend, and the average value finally stabilized at approximately 1.5 s.

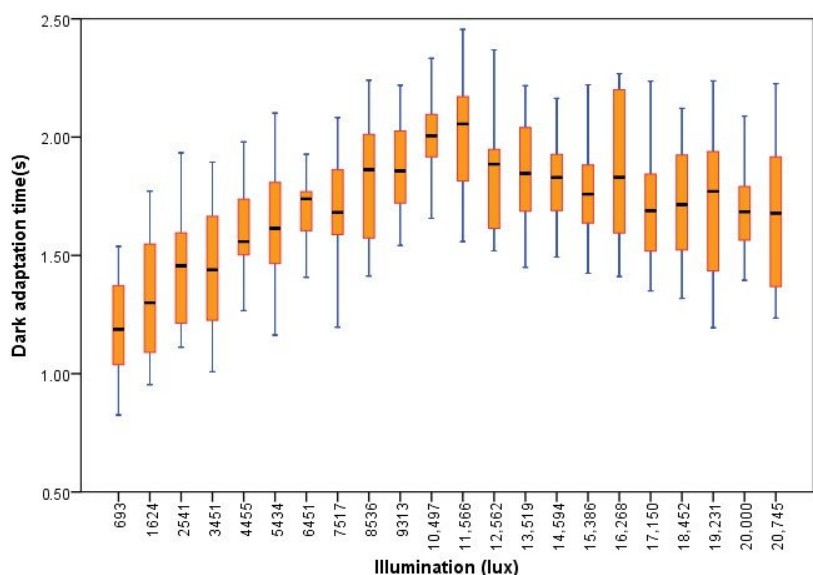

**Figure 16.** Relationship between the dark adaptation time and illuminance.

Figure 17 shows that light adaptation time increased with increasing illuminance and reached a maximum value from 12,000 to 13,000 lux. When the illuminance was higher

than 12,000–13,000 lux, the light adaptation time gradually decreased with increasing illuminance, but the downward trend was slower than the upward trend, and the average value finally stabilized at approximately 0.9 s.

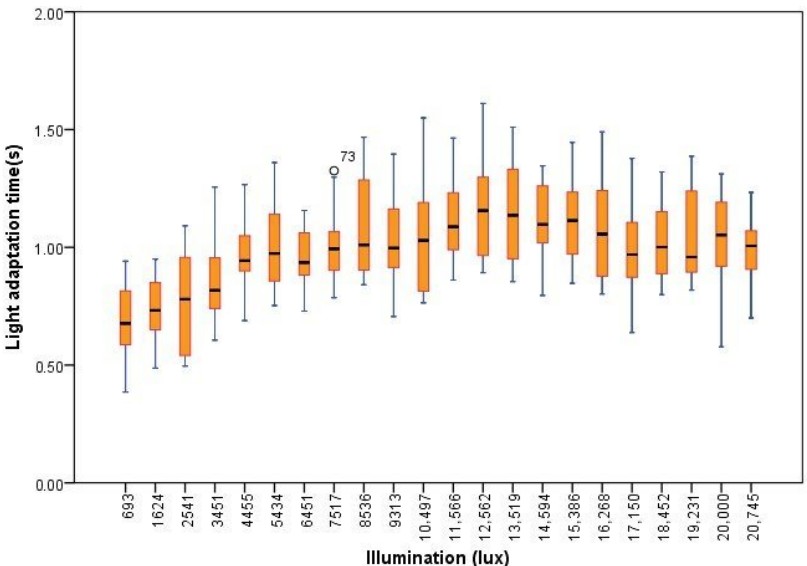

**Figure 17.** Relationship between the light adaptation time and illuminance.

### 3.4.2. Establishing the Relationship between Adaptation Time and Illuminance

Based on the average adaptation time at each illuminance level in the Experiment 2 data, we analyzed the relationship between the average adaptation time and illuminance and determined the illuminance–average adaptation time relation via fitting, as shown in Figure 18.

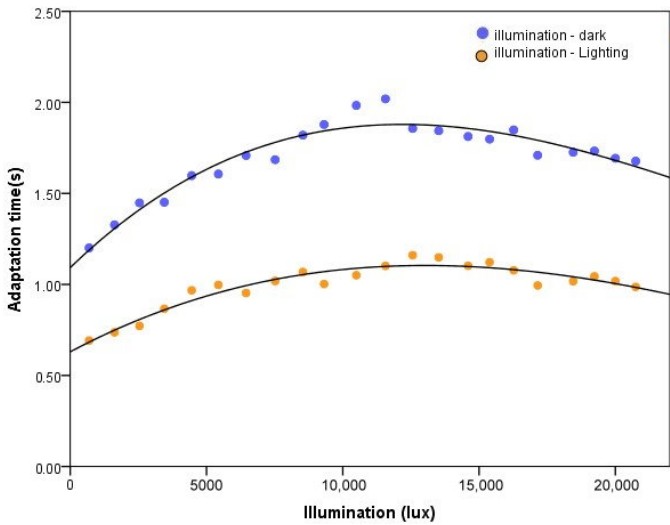

**Figure 18.** Adaptation time and illuminance model.

Equations (3) and (4) were obtained by establishing the relationships between the average adaptation times and illuminance.

$$T_1 = 1.03 \times 10^{-13} l^3 - 7.81 \times 10^{-9} l^2 + 1.44 \times 10^{-4} l + 1.09, \qquad (3)$$

$$T_2 = 3.74 \times 10^{-14} l^3 - 3.77 \times 10^{-9} l^2 + 7.91 \times 10^{-5} l + 0.63, \qquad (4)$$

where $T_1$ = average dark adaptation time; $T_2$ = average light adaptation time; $l$ = illuminance.

To validate the model, we performed a correlation analysis and *F* tests on Equations (3) and (4), and the results are summarized in Tables 5 and 6, respectively.

**Table 5.** Model correlation test.

| Type | $R$ | $R^2$ | Adjusted $R^2$ | Standard Error |
|---|---|---|---|---|
| Dark adaptation | 0.960 | 0.922 | 0.909 | 0.061 |
| Light adaptation | 0.958 | 0.919 | 0.905 | 0.039 |

**Table 6.** Model *t*-test.

| Type | | Sum of Squares | df | Mean Squared Error | F | Significance *p* |
|---|---|---|---|---|---|---|
| Light adaptation time | Between groups | 0.787 | 3 | 0.262 | 70.923 | ≤ 0.001 |
| | Within groups | 0.067 | 18 | 0.004 | | |
| | Total | 0.854 | 21 | | | |
| Dark adaptation time | Between groups | 0.309 | 3 | 0.103 | 67.756 | ≤ 0.001 |
| | Within groups | 0.027 | 18 | 0.002 | | |
| | Total | 0.337 | 21 | | | |

Table 5 indicates that $R^2 > 0.8$, which implies that the degree of model fitting is acceptable.

Table 6 indicates that the significance value $p \leq 0.001$, demonstrating that there is a significant cubic function relationship between adaptation time and illuminance.

In summary, we have demonstrated the following:

- Figure 18 shows that at the same illuminance, the dark adaptation time is longer than the light adaptation time, which is consistent with the previous analysis results.
- Illuminance directly affects the dark and light adaptation times, and they generally have cubic relations.
- Equations (3) and (4) confirm statistical significance.

## 4. Conclusions and Future Work

In this paper, we have analyzed the relationships between light and dark adaptation times and age, gender, and illuminance in simulation experiments. We also studied the visual light and dark response characteristics of drivers at tunnel entrances. The following conclusions can be drawn.

- After a driver enters or exits a tunnel, a certain vision adaptation time is required, because of drastic changes in the light environment. This is because the pupil changes the intensity of the light [9]: when entering the tunnel, the light will darken, the pupil will enlarge, and the eye will see black; when leaving the tunnel, the light will become bright, the pupil will contract and the eye will see white. This is also the reason why there are more accidents at tunnel entrances than on ordinary road sections [8].
- The experimental results indicate that the dark and light adaption times ranged from 1.2 to 2.4 s and 0.7 to 1.5 s, respectively. The adaptation times for dark and light were slightly longer than those of 2.3 and 1.3 s in [10], but the conclusions were similar overall. Under all illuminance conditions, the dark adaptation time was longer than the light adaptation time, indicating that the visual discomfort for drivers caused by a tunnel entrance is more severe than that caused by a tunnel exit, i.e., the black hole effect is more dangerous than the white hole effect. With increasing age, the difference between the light and dark adaptation times increases.
- Experiment 1 showed that there was a significant positive correlation between dark adaptation time and age; namely, with increasing driver age, the dark adaptation time increased, while the correlation between light adaptation time and age was not

significant. The main reason for this [4,14] is that the elderly need a longer fixation time, and the efficiency of visual information processing is reduced.

- Under different age and illuminance conditions, the adaptation time remained generally the same regardless of gender. It can be concluded that there is no correlation between gender and light and dark adaptation times.
- Dark adaptation time was highly correlated with illuminance. Light and dark adaptation times increased with increasing illuminance, reaching maximum values from 12,000 to 13,000 lux and from 11,000 to 12,000 lux, respectively. When the illuminance continued to increase, the adaptation time gradually decreased with increasing illuminance, but the downward trend was slower than the upward trend.

Our ultimate goal is to digitize the light and dark vision response characteristics of drivers in tunnels, e.g., the adaptation time in tunnel portals in different directions, to provide a broader theoretical basis for tunnel portal linear lighting design.

Therefore, in future work, we will expand our research. First, the number of samples across different age groups will be increased, and the sample data accuracy will be improved. Second, the difference in illuminance levels will be increased, and the laws of light and dark vision adaptation characteristics will be studied for larger illuminance differences. Third, an appropriate test scheme for a tunnel mouth will be established.

**Author Contributions:** Conceptualization, C.L.; Methodology, C.L.; Software, C.L.; Validation, C.L. and Q.W.; Formal Analysis, C.L.; Investigation, C.L. and Q.W.; Resources, C.L.; Data Curation, C.L.; Writing—Original Draft Preparation, C.L.; Writing—Review and Editing, C.L. and Q.W.; Visualization, Q.W.; Supervision, Q.W.; Project Administration, C.L.; Funding Acquisition, C.L. All authors have read and agreed to the published version of the manuscript.

**Funding:** This research was funded by Shaanxi Province Transport Planning Design and Research Institute Fund, grant number 20-19.

**Institutional Review Board Statement:** The study was conducted according to the guidelines of the Declaration of Helsinki, and approved by the Institutional Review Board of Shaanxi Province Transport Planning Design and Research Institute (protocol code 2020-06 and 27 April 2020).

**Informed Consent Statement:** Written informed consent has been obtained from the patient(s) to publish this paper.

**Data Availability Statement:** Restrictions apply to the availability of these data. Data was obtained from Shaanxi Province Transport Planning and Research Institute and are available from the authors with the permission of Shaanxi Province Transport Planning Design and Research Institute.

**Acknowledgments:** The authors gratefully acknowledge Shaowei Yang and Binghong Pan for useful discussions and consultations.

**Conflicts of Interest:** The authors declare no conflict of interest. The funder had no role in the collection of data, and no role in the design of the study; in the collection, analyses, or interpretation of data; in the writing of the manuscript, or in the decision to publish the results.

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
