# Peer review of "Simulating Human Visual Perception in Tunnel Portals"

_sustainability, doi:10.3390/su13073741_

Round 1

Reviewer 1 Report

The article is interesting. However, some aspects should be taken into account for improving it.

Despite I am not an English native in my opinion English writing should be improved, especially in relation to scientific stile of the text and grammar.

Moreover, the text presented seem to be a draft due to some words appeared highlighted in yellow. There are also some mistakes in some of figures presented, as I will detail bellow. So please review if this document is really your final version.

Specifics comments:

Line 25: Summary is not the adequate word. Much better use “Introduction”.

Additionally, in this section aims of the work should be better detailed.

Line 91: “Experimental scheme” should be better changed by “Material and methods”.

Line 96: [error! Reference source not found] Please correct it.

2.5 Experimental design

Why did you decided eliminate in the selection of experiment 2 groups 51-60 and 61-70? It is true that you have less participants in both groups but you only need 4 males and females of each one. Please justify.

Line 162: “Results and discussion” much better than results and analysis.

First of all, results should be discussed by the support of scientific citations, but I have no found any in this section, so this part of the paper should be redone.

Figures 4 and 5 are the same graphic! Figure 5 should represent relationship between the adaptation time difference and illuminance…and both graphic represent “age” in axis X.

Moreover, you should reference figures in the text when you describe them and sometimes you do not do it, e.g., you don't quote figure 6 in line 168.

Again figures 6 and 7 are the same graphic! Figure 7 isn`t correct. To this respect, I haven’t been able to interpret the analysis exposed between lines 182-185.

3.2. Characteristics of light and dark adaptation based on age

Line 189-190: …”we selected the data at an illuminance ranging from 500-8000 lux for a total of 20 effective samples”. What do “samples” really mean? 20 people? Clarify at this point what “samples” means, and what is the number of total participant selected in this analysis.

Table 1: N=187. Clarify also the sample size of this analysis.

3.3 Characteristics of light and dark adaptation based on gender

Line 242: The adjusted effective data contained 192 groups. Please Define “groups” at this point. ¿192 people from the total of 237?

3.4 Characteristics of light and dark adaptation based on Illuminance

Line 259: “..effective sample data consisted of 176 groups”. Again explime and define much better what this number really represents from the whole of 24 people selected in experiment 2.

Conclusion and Future Work

At this point I have a question for you:

How do you consider speed of driving can affect to light time adaptation which is analysed in your work? Your experiment is based on people walking through a light-shielded tent, but if you want to extrapolate results to the field of drivers may be you should contemplate speed as a variable to be study in the future. Have you considered that?

Author Response

Dear Ms. Rudy Miao and Reviewers,

Thanks very much for taking your time to review this manuscript. I really appreciate all your comments and suggestions! Please find my itemized responses to below and my revisions/corrections in the re-submitted files.

Best regards,

<Changjiang Liu>

<Qiuping Wang>

Reviewer 2 Report

The manuscript reports the findings of an empirical investigation focused on the light-dark adaption in a drivable tunnel. In this study, the relationship between age, gender, illuminance and light and dark adaptation times was examined, and a model was suggested subsequently.

The study is relevant, deals with a sustainable topic and could enrich the ongoing debates in this field. However, there are a couple of aspects which the authors should consider in the preparation of a revised version of the manuscript:

  • In your study, you define different light conditions (and lux values) as study parameters. Are these values comparable to other projects, or can they be derived or reasoned / justified with references to state-of-the-art literature?

  • The actual tunnel is, of course, a real-world case study. However, it is hardly possible to avoid or even minimize disturbing effects between the tests of the different participants. Therefore, researchers dealing with virtual environments (featuring high levels of immersion and realism) have suggested to use these simulations as alternative labs (with reduced disturbances). It would be worth mentioning examples of this alternative study scenario in an outlook, or in the introduction. These recent papers could serve as examples:

Lokka, I.E., Çöltekin, A., (2020). Perspective switch and spatial knowledge acquisition: effects of age, mental rotation ability and visuospatial memory capacity on route learning in virtual environments with different levels of realism. In: Cartography and Geographic Information Science, 47(1), 14-27. https://doi.org/10.1080/15230406.2019.1595151

Keil, J., Edler, D., O'Meara, D., Korte, A., Dickmann, F.  (2021). Effects of Virtual Reality Locomotion Techniques on Distance Estimations. In: ISPRS International Journal of Geo-Information, 10 (3): 150. https://doi.org/10.3390/ijgi10030150

  • In your conclusion, you list up several findings which are connected to your study results. The listing up of aspects is a quite unusual way, but effective (and thus acceptable). So far, there is however no connection between your study results and other related study. While your background section includes many citations of related studies, you do not refer back to them here. This linkage is important in a final version, and it would increase the value of your study.

  • A minor issue: Section 2.1, includes a formatting feedback at one stage (l. 96): “Error! Reference source not found”. Would you please add the correct reference?

Author Response

(The authors gave the same response as above.)

Round 2

Reviewer 1 Report

Once I have checked the modifications that authors have made and their justified replies.

Reviewer 2 Report

The authors provided a revised version of the manuscript. This version includes changes considering the points addressed in review round no. 1. I am a bit surprised about the short conclusion, however with the keypoint listed. The response letter is also a bit unusual, as it points to changes (rather techically). But it does not show the thoughts behind the changes made. In total, the manuscript convinces through its focusedness.